# Pathophysiological Mechanisms and Treatment of Dermatomyositis and Immune Mediated Necrotizing Myopathies: A Focused Review

**DOI:** 10.3390/ijms23084301

**Published:** 2022-04-13

**Authors:** Renske G. Kamperman, Anneke J. van der Kooi, Marianne de Visser, Eleonora Aronica, Joost Raaphorst

**Affiliations:** 1Department of Neurology, Amsterdam UMC, University of Amsterdam, Amsterdam Neuroscience, 1100 DD Amsterdam, The Netherlands; a.j.kooi@amsterdamumc.nl (A.J.v.d.K.); m.devisser@amsterdamumc.nl (M.d.V.); j.raaphorst@amsterdamumc.nl (J.R.); 2Department of (Neuro)Pathology, Amsterdam UMC, University of Amsterdam, Meibergdreef 9, 1105 AZ Amsterdam, The Netherlands; e.aronica@amsterdamumc.nl

**Keywords:** myositis, pathophysiological mechanisms, interferon pathway, treatment, DM, IMNM

## Abstract

Idiopathic inflammatory myopathies (IIM), collectively known as myositis, are a composite group of rare autoimmune diseases affecting mostly skeletal muscle, although other organs or tissues may also be involved. The main clinical feature of myositis is subacute, progressive, symmetrical muscle weakness in the proximal arms and legs, whereas subtypes of myositis may also present with extramuscular features, such as skin involvement, arthritis or interstitial lung disease (ILD). Established subgroups of IIM include dermatomyositis (DM), immune-mediated necrotizing myopathy (IMNM), anti-synthetase syndrome (ASyS), overlap myositis (OM) and inclusion body myositis (IBM). Although these subgroups have overlapping clinical features, the widespread variation in the clinical manifestations of IIM suggests different pathophysiological mechanisms. Various components of the immune system are known to be important immunopathogenic pathways in IIM, although the exact pathophysiological mechanisms causing the muscle damage remain unknown. Current treatment, which consists of glucocorticoids and other immunosuppressive or immunomodulating agents, often fails to achieve a sustained beneficial response and is associated with various adverse effects. New therapeutic targets have been identified that may improve outcomes in patients with IIM. A better understanding of the overlapping and diverging pathophysiological mechanisms of the major subgroups of myositis is needed to optimize treatment. The aim of this review is to report on recent advancements regarding DM and IMNM.

## 1. Introduction

Idiopathic inflammatory myopathies (IIM), collectively known as myositis, are a composite group of rare autoimmune diseases affecting mostly skeletal muscle, although other organs or tissues may also be involved [1]. The current classification includes dermatomyositis (DM), immune-mediated necrotizing myopathy (IMNM), anti-synthetase syndrome (ASyS), overlap myositis (OM) and inclusion body myositis (IBM) [1,2,3]. Polymyositis is a contested entity within the spectrum and is considered a diagnosis by exclusion [4,5,6]. Clinical features may vary between and within subgroups. DM is characterized by skin involvement; other extramuscular features are frequently encountered, such as interstitial lung disease (ILD) [2]. IMNM and IBM are the most ‘muscle predominant’ phenotypes within the IIM spectrum [7,8]. In ASyS, multisystem involvement is frequent, i.e., ILD and arthritis are often the presenting symptoms and (peri)myocarditis may occur [9]. Overlap myositis is often associated with connective tissue disorders [10,11].

In recent decades, both adaptive and innate immune mechanisms have been shown to contribute to the pathogenesis of IIM. The characterization of T and B cell populations in inflammatory infiltrates, the discovery of myositis-specific autoantibodies (MSA) and myositis-associated autoantibodies (MAA) with corresponding clinical phenotypes, and new insights into the role of derangements of the complement system and the important contribution of interferonopathy have led to increased understanding and new targets for therapy.

A better understanding of the overlapping and diverging pathophysiological mechanisms of the major subgroups of myositis is needed to optimize individual treatments for adult myositis patients.

This review aims to describe current knowledge on pathophysiological mechanisms in myositis, in relation to current and future pharmacological treatments.

We focus on two of the most prevalent subtypes of IIM in adults: dermatomyositis (DM) and immune-mediated necrotizing myopathy (IMNM). Other subtypes show a considerable overlap with rheumatic disorders (Table 1); for these subtypes we refer to focused reviews [12,13]. IBM usually presents with a gradual onset and is slowly progressive over years, characterized by asymmetric weakness [14,15]. Along with inflammatory findings, degenerative disease mechanisms play an important role in IBM, although the etiology of inflammation and degeneration is so far unresolved [16,17]. IBM tends to be treatment-refractory. As mentioned above, this review focuses on IIM in adults; a review focusing on juvenile DM was published in 2017 [18].

For DM and IMNM, we first describe the phenotype, including findings in muscle biopsies, and subsequently we describe three pathophysiological mechanisms: the humoral immune response including the role of antibodies, derangements of the complement system, and overexpression of the interferon pathway. In addition, we review treatment options with an emphasis on these three pathophysiological mechanisms.

### 1.1. Clinical Features

Dermatomyositis (DM) is the most common (~30–40%) subtype of IIM [2,3,21]. DM is characterized by predominant proximal muscle weakness in combination with a typical skin rash, in particular Gottron papules/sign or heliotrope erythema (Table 1). Muscle weakness leads to problems with swallowing, running, climbing stairs, rising from the floor, and/or raising arms [2]. On laboratory testing, serum creatine kinase (CK) activity is often elevated and, in addition, myositis specific antibodies (MDA-5, NXP-2, Mi-2, TIF1-γ, SAE-1) can be found in ~60% of patients [22]. The association with cancer is an important feature of DM: approximately 8% of patients with DM have or develop cancer (standardised incidence ratio 2.4–6.2) [23,24].

Immune mediated necrotizing myopathy (IMNM) is the second most common subtype of IIM (~20%) [8,21]. IMNM is characterized by rapid-onset severe proximal muscle weakness without clinically significant extramuscular disease activity. Serum CK activity is often highly elevated. Two autoantibodies are associated with IMNM: anti-signal recognition particle (SRP) and anti-3-hydroxy-3-methylglutaryl-CoA reductase (HMGCR) autoantibodies. There is also a seronegative subtype in which these MSAs cannot be identified; each of these three IMNM subtypes (SRP+, HMGCR+ and seronegative) constitute one-third of patients with IMNM. In patients with anti-HMGCR autoantibodies, a history of statin use is found in approximately half of the patients [25]. Seronegative and anti-HMGCR-positive IMNM are associated with cancer, while anti-SRP-positive IMNM is not [26,27].

### 1.2. Muscle Biopsy

In dermatomyositis, muscle biopsies show inflammatory infiltrates predominantly perivascularly, in the interfascicular septae and/or at the periphery of the fascicles (Figure 1). The infiltrate consists of B cells, macrophages, dendritic cells and CD4 T cells. Muscle fibers show atrophy at the periphery of the fascicle, the characteristic perifascicular atrophy (PFA). Unlike other subtypes, prominent necrosis is less frequently observed in DM biopsies. The presence of PFA has a high specificity (>90%), but sensitivity is restricted (~50%), primarily because PFA is distributed in a patchy manner [28,29]. These fibers are COX-deficient fibers and/or positive with CD56 (neural cell adhesion molecule, NCAM) staining and consistently show overexpression of perifascicular myxovirus resistant A (MxA), a type I interferon (IFN-1)-induced protein (Figure 1) [2,30,31]. Expression of MxA in myofibers is highly specific in DM, suggesting it plays an important role in the pathophysiology of DM [32,33,34]. The cause of pathological alterations in the perifascicular muscle fibers is still not entirely clear: they could be due to expression of IFN-1-induced gene products, or ischemia because of capillary depletion, or a mixture of both [35,36,37].

Histopathological evidence of a microvascular pathogenesis in DM is further suggested by capillary abnormalities which were suggested to precede muscle fiber damage and other structural changes: the characteristic finding early in the disease process is the presence of microtubular inclusions in endothelial cells, often preceding inflammatory cell infiltrates [38,39]. These inclusions are related to the endoplasmatic reticulum (ER) or to the outer nuclear membrane, and probably represent membranous specializations within the ER during a certain stage of cellular activity [40]. MAC deposition in capillary walls on endomysial capillaries has a high sensitivity and specificity in the diagnosis of DM, differentiating it from other subtypes of IIM [41].

In IMNM, muscle biopsies show abundant necrotic fibers invaded or surrounded by macrophages in combination with regeneration of muscle (Figure 1). There is variable expression of major histocompatibility complex type I (MHC-I) in non-necrotic fibers and diffuse and fine expression of sarcoplasmic p62 [34,42]. The macrophages in inflammatory infiltrates in IMNM probably play a role in tissue repair [43,44]. This may explain why, in muscle biopsies of IMNM, a higher frequency of regenerating fibers compared to necrotic fibers is seen, when compared to DM biopsies [45].

Although the proportion of necrotic fibers correlates with CD3 and CD8 cells and with the proportion of MHC-I positive fibers, no signs of overt cytotoxicity are typically detected in IMNM biopsies. Invasion of non-necrotic muscle fibers by cytotoxic CD8+ T cells (which is characteristic for inclusion body myositis) was not observed in anti-SRP positive and anti-HMGCR positive muscles, and the surrounding of muscle fibers by T cells was a rare event [45].

IMNM muscle biopsies characteristically show sarcolemmal and sarcoplasmic deposition of complement (Membrane Attack Complex (MAC) on necrotic fibers (Figure 1). Correlation between sarcolemmal deposits of C5b-9 and fiber necrosis has been shown [45].

## 2. Pathophysiological Mechanisms

The exact pathophysiological mechanisms causing IIM remain unknown. In general, and similarly to most other auto-immune disorders, an exogenous trigger (e.g., a viral infection) is presumed to be required in combination with a genetic predisposition. In some patients, a clear trigger causing IIM can be identified. Anti-HMGCR is, in up to 67% of cases, associated with prior statin use, particularly in patients older than 50 years of age [8]. DM can occur as a paraneoplastic syndrome [46].

Recently, several cases of IIM following SARS-CoV-2 vaccination or COVID-19 infection have been reported, including patients with myositis-specific antibodies [47]. In a case-control study, the autopsy of most patients who died from severe COVID-19 showed signs of myositis ranging from mild to severe inflammation, with sarcolemmal major histocompatibility class 1 and in some cases class 2 staining patterns. In some patients, evidence of capillary interferon overexpression and complement activation was noted, suggesting type-1 interferonopathy. The authors did not demonstrate direct viral infection of myofibers [48]. This led to the suggestion that SARS-CoV-2 may be associated with postinfectious myositis.

In IIM, similarly to other autoimmune and immune-mediated diseases, the strongest genetic risk for disease susceptibility is presumed to localize to specific leucocyte antigen (HLA) alleles [49,50]. The strongest associations have been found by stratifying patient groups according to antibody status (see section below) [51]. Unique immunogenetic backgrounds were found in Asian IIM patients, when compared to Caucasian populations [52].

Immune-mediated disease mechanisms are thought to encompass both adaptive and innate disease mechanisms [53]. Previous reports postulated a prominent role for B cells in DM and for cytotoxic CD8+ T-cells in other IIM subtypes [54]. More recent reports indicate that cellular adaptive immunity, humoral adaptive immunity, and innate immunity are likely all of importance in the pathogenesis of all subtypes [53]. In short, the pathogenicity of at least some myositis-specific autoantibodies is plausible, as supported by the finding of related auto-antigen overexpression in IIM patients (e.g., TIF1-γ overexpression in endometrial cancer cells in patients with cancer-associated DM with anti-TIF1-γ autoantibodies) and experimental passive immunization via auto-antibody transfer (in in vitro and in vivo models of seropositive IMNM) [26,55,56,57,58]. The muscle microenvironment in IIM is enriched with both myeloid and plasmacytoid antigen-presenting/dendritic cells, and muscle and muscle-infiltrating cells overexpress innate immune receptors, including Toll-like receptors, which may lead to nuclear factor kappa B (NF-κB) and pro-inflammatory cytokine and chemokine signaling. This leads to IFN-1 overexpression in dermatomyositis, which may cause a pro-inflammatory state, MHC-class I overexpression, and non-immune mediated toxicity via mitochondrial damage and endoplasmatic reticulum stress [30,59].

Non-immune-mediated factors may also contribute to pathogenesis: muscle hypoxia, reactive oxygen species, heat shock response, endoplasmic reticulum stress, and abnormalities of cellular autophagy have all been described in IIM [60,61]. In short, these mechanisms create a positive feedback loop with inflammation, leading to subsequent impaired muscle contraction, muscle protein dyshomeostasis, and disease damage with atrophy and irreversible muscle fiber alterations [61,62].

Various disease mechanisms have been postulated in IIM, of which we discuss in depth the role of the humoral immune response (B cells and antibodies), derangements of the complement system, and overexpression of the interferon pathway, in the most prevalent subtypes of IIM, dermatomyositis (DM) and immune mediated necrotizing myopathy (IMNM).

## 3. Humoral Immunity in DM and IMNM: The Role of B Cells and Antibodies

### 3.1. B Cells

Humoral immunity is a specific immune response mediated by antibodies. Sixty to seventy per cent of IIM patients have myositis-specific (~50%) or myositis-associated (~20%) antibodies [63,64]. These antibodies are produced by plasma cells and, to a lesser extent, plasmablasts. The latter differentiate from antigen-specific B cells, which are present in very low frequencies in peripheral blood. A crucial role of B cells in the pathogenesis of IIM is evidenced by the relative effectiveness of B cell directed therapies such as rituximab in the treatment of IIMs [65,66,67], the endomysial production of B cell activating factor (BAFF), the presence of B cells and plasma cells in muscle tissue, antibody production, and endomysial B cell-maturation in part within the muscle itself [68,69].

#### 3.1.1. B Cells in DM

DM muscle biopsies are especially known to include infiltrating CD20+ B cells, predominantly in a perivascular distribution within the perimysium [44,70]. The number of inflammatory cells varies from sparse infiltrates to B-cell rich infiltrates to nodular, follicle-like collections. This so-called ectopic lymphoid neogenesis, which can be seen in nonlymphoid tissue in different autoimmune diseases, is thought to be extremely rare in DM [69,71]. Radke et al. [69] demonstrated that IFN-1 signature related gene-expression levels paralleled B cell content and architectural organization and linked B cell immunity to the IFN-1 signature. They hypothesized that in situ B cell differentiation in skeletal muscle in DM may dysregulate immunity, leading to excessive IFN response, which probably fuels an amplification loop [69].

The role of antigen-driven B cell adaptive immune responses within the inflamed muscle of IIM patients are not fully understood yet. Recent data suggest that variability, diversity, and joining (VDJ) repertoires of B cells from muscle tissues of DM patients deviate from normal immunoglobulin heavy chain (IgH) gene repertoires [72]. These VDJ repertoires of muscle infiltrating B cells of DM patients often undergo clonal diversification and accumulate high somatic hypermutations, suggesting an antigen-driven response as well as a possible role of B cells in the pathological mechanisms responsible for this disorder.

#### 3.1.2. B Cells in IMNM

In IMNM muscle biopsies, B cell mediated pathology is not as prominent when compared to DM biopsies; in both anti-SRP positive and anti-HMGCR positive muscles, CD79+ B cell density was low, as was CD138+ plasma cell density [45]. This in agreement with others reporting anti-HMGCR subjects showing small numbers of CD20+ cells, which were restricted to the endoymysium [44]. On the other hand, an important role of BAFF in the pathological mechanism of IMNM has been suggested: in refractory patients with anti-SRP positive IMNM, BAFF-receptor (BAFF-R) expression in muscle is upregulated when compared to non-refractory patients. This finding implicates BAFF as a possible contributor to muscle fiber injury in IMNM [73]. Additionally, autoantibodies in IMNM patients (anti-SRP and anti-HMGCR antibodies) have been reported to be crucial in initiating immune events, which are described below in more detail [73,74].

### 3.2. Autoantibodies

In the last decade over 30 different myositis-specific antibodies (MSA) and myositis-associated antibodies (MAA) have been identified. Out of these, 14 MSAs and 6 MAAs help discriminate between subgroups and predict a higher chance of malignancy (anti-TIF1-γ, NXP-2, HMGCR) and/or extramuscular manifestations (e.g., ILD—MDA-5), and are therefore of prognostic importance [75,76]. Regarding associations with autoantibodies and malignancies, no dominating type of malignancy was observed in association with DM or IMNM [77]. One hypothesis regarding the potential mechanisms of cancer-associated myositis (CAM) is shared antigen expression [78]. In patients with DM, shared expression of several antigens is observed between tumor and regenerating muscle cells, indicating that an autoimmune response against cancer could cross-react with regenerating muscle tissue, leading to auto-immunity [79]. The exact pathophysiological mechanism regarding the presence of myositis specific autoantibodies and development of cancer remains as yet unknown.

The pathogenic role of the MSAs, if any, is unknown for most of the antibodies. For most of the MSAs it remains to be determined whether the antigens reside on the surface of muscle cells (sarcolemma) in order to be targeted by the antibody, and if not, how antigens are reached intracellularly by the antibodies. Similarly, it is not known for most MSAs whether they might cross-react with a different, unidentified antigen and whether the autoantibodies, other soluble factors, or infiltrating immune cells cause the muscle damage [80]. Below we focus on two MSAs in DM, i.e., anti–transcription intermediary factor 1-γ (TIF1-γ) and anti–melanoma differentiation gene 5 (MDA-5); and two MSAs associated with IMNM, i.e., SRP and HMGCR. For these four MSAs a few lines of evidence have been proposed which may (partly) explain their role in the pathogenicity of DM and IMNM.

#### 3.2.1. Autoantibodies in DM

DM is associated with five MSAs, i.e., anti-TIF1-γ, anti-MDA-5, anti-SAE-1, anti-Mi-2 and anti-NXP-2 (Table 2). The discovery of dermatomyositis-specific antibodies (DMSAs) resulted in valuable insights into the distinct features associated with different antibodies [32]. Relative proportions of DMSAs may differ between countries and continents. In a nationwide study on the prevalence of MSAs in the Netherlands (Euroline myositis line-blot assay; Euroimmun), out of 88 patients with an MSA, 49% of MSAs were DMSAs, with relative proportions ranging between 15% (anti-TIF1-γ) and 2% (anti-SAE-1) [81]. Different DMSA-associated clinical phenotypes have been defined; in anti-TIF1-γ DM, skin lesions, dysphagia, and malignancy are frequent [82,83,84], whereas ulcerating skin lesions, interstitial lung disease (ILD) with low CK activity, and less frequent muscle involvement are encountered in anti-MDA-5 DM [85,86].

Related to the focused nature of the review we discuss two of the five DMSAs (MDA-5 and TIF1-γ) in the following sections. This is based on literature, albeit scarce, suggesting a pathogenic role (TiF1-y) or a link with an important immunological pathway in dermatomyositis (interferon overexpression; anti-MDA-5).

##### Anti-MDA-5 DM

Anti-MDA-5 antibodies are one of the five currently recognized antibodies associated with DM.

MDA-5 was identified in patients with clinically amyopathic DM (CADM). Anti-MDA-5 positive patients are characterized by a skin rash, often with signs of skin vasculopathy and ulcerating skin lesions (typically at the palmar surface of the hands), polyarthritis and rapid ILD with a high mortality rate (between 33–67%) [85,87,88,89,90,91]. Clinical signs of myositis are usually absent or mild [92]. The vast majority of studies on anti-MDA-5 DM are from Asia [93,94]. Anti-MDA-5 antibodies titers correlate with the severity of disease, and higher levels are associated with rapidly progressive ILD in juvenile DM [95,96]. MDA-5 is known to be a cytosolic protein that operates as a virus RNA sensor and induces the production of IFN-1 and pro-inflammatory cytokines by the cell [87].

Since IFN-1 signaling is elevated both in skin and serum of patients with anti-MDA-5 DM, and the IFN-1 signaling in this DM type exceeds that in other types, such as anti-MDA-5 negative patients, the existence of a specific trigger for the IFN-1 pathway in MDA-5 is suggested [97,98,99]. Anti-MDA-5 antibodies could be essential to the dysregulation of the IFN-1 pathway and a few possible contributions of anti-MDA-5 antibodies to pathogenesis have been postulated [87]. Interaction between anti-MDA-5 antibodies and an ectopic antigenic target could trigger activation of the IFN-1 signaling pathway, along with immune mediated cytotoxicity through complement activation and antibody-dependent cytotoxicity [87]. MDA-5 released from apoptotic cells or from skin or lung fibroblasts could also bind to anti-MDA-5 antibodies, forming immune complexes. According to Nombel et al., these immune complexes could potentially deposit in organs, e.g., in dermal or pulmonary vasculature, inducing vascular damage [87].

Finally, anti-MDA-5 antibodies may have the ability to penetrate cells and interact with cytoplasmic MDA-5, similarly to other antibodies, altering various functional pathways [87,100,101,102]. These potential contributions of MDA-5 antibodies to the pathogenesis of DM should be further investigated.

##### Anti-TIF1-γ DM

Anti-TIF1-γ antibodies are one of the five currently recognized antibodies associated with DM. In a murine model, the immune response against TIF-1γ seems to mediate the induction of myositis [103]. However, the same authors showed that B cells and autoantibodies are not necessary for the development of TIF1-γ-induced murine myositis; therefore, the development of anti-TIF1γ autoantibodies cannot be linked directly to a pathogenic role [103].

In adults only, TIF1-y antibodies are strongly associated with an underlying malignancy. Approximately 60–80% of anti-TIF1γ-positive patients have cancer-associated IIM [76]. Gastric, lung, breast, esophageal, bladder, colorectal, ovary, and thymus tumors are found. Of interest, anti-TIF1-γ DM is also associated with malignancy in younger patients (<40 years) [32].

Anti-TIF1-γ plays a role in tissue differentiation, DNA repair, and tumor suppression [76]. The pathogenicity of anti-TIF1-γ is plausible, as supported by the finding of TIF1-γ overexpression in endometrial cancer cells in patients with cancer-associated DM with anti-TIF1-γ autoantibodies [104]. A significantly higher expression of TIF1-γ has been reported in tumors and muscles of anti-TIF1-γ positive DM patients as compared to non-TIF1-γ associated DM patients [105].

#### 3.2.2. Autoanntibodies in IMNM

IMNM is associated with two specific autoantibodies: anti-SRP and anti-HMGCR [106,107]. An increased incidence of cancer is seen in patients with anti-HMGCR antibodies and in seronegative IMNM patients [108]. As described above, in IMNM, the sparse inflammatory infiltrates and presence of complement deposits on the sarcolemma of myofibers may point to an auto-antibody-mediated pathophysiology rather than T cell-dependent cytotoxicity. However, since the expression of the auto-antigens SRP and HMGCR is ubiquitous rather than muscle-specific, the pathogenic role of autoantibodies directed against these molecules in IMNM has remained elusive until the last five years [77,108,109].

##### Anti-SRP IMNM

In the abovementioned nationwide study on the prevalence of MSAs in the Netherlands, out of 88 patients with an MSA, 13% of patients had anti-SRP antibodies [81]. Compared to other IIMs, anti-SRP myopathy is associated with more pronounced clinical weakness [45,110,111], highly elevated serum CK levels [45], extensive muscle edema with early fatty degeneration on muscle MRI [112], and abundant necrotic fibers on biopsy [45].

The course of the disease tends to be rapidly progressive with severe disability within months [113]. In a longitudinal study of IMNM patients, anti-SRP antibody titer correlated with serum CK activity and muscle strength [114]. Recent in vitro and in vivo studies have increased our understanding of the pathogenic role of anti-SRP autoantibodies in IMNM.

SRP is a ribonucleoprotein (RNP) that mediates nascent protein translocation to the endoplasmic reticulum (ER) [115,116]. Anti-SRP autoantibodies have been shown to inhibit SRP function in vitro [117]. Muscle co-culture with anti-SRP positive serum has been shown to reduce myoblast viability, more markedly upon addition of complement [118]. In another in vitro study, anti-SRP antibodies induced myotubular atrophy, increased pro-inflammatory cytokines, and impaired myoblast fusion due to decreased IL-4/IL-13 [55].

In an in vivo murine model, anti-SRP IgG passive transfer led to myofiber necrosis and muscle deficit, which was less evident in C3-deficient mice; active immunization with anti-SRP antibodies elicited muscle weakness [119]. The mechanism by which antibodies may interact with their cytoplasmic, ubiquitous cognate antigens and selectively induce muscle pathology is as yet unresolved. Antibodies could block SRP function intracellularly; capacity for cell penetration has been confirmed for other antibodies, such as anti-DNA [120,121,122,123] and anti-RNP [124] antibodies.

As opposed to the intracellular effects of anti-SRP antibodies, sarcolemmal SRP expression and consequential classical complement pathway activation poses another pathogenic pathway. For other ER chaperones, cell surface expression has been observed in various cells [125] and in specific circumstances, such as macrophage fusion [126]. Indeed, studies reported sarcolemmal SRP expression through co-labeling with membrane-associated proteins (dysferlin and dystrophin). Immunostaining of IMNM muscle biopsies of patients with SRP antibodies also revealed that SRP positive fibers were partially co-labeling with sarcoplasmic neural cell adhesion molecule (NCAM), a marker of regeneration [45]. Co-localization of complement (C3c) with sarcolemmal and sarcoplasmic SRP expression has been shown [45,118].

In summary, current evidence implies pathogenicity of the SRP-specific immune response, possibly through multiple effector functions including complement-dependent antibody-mediated cytotoxicity. The exact immune-mediated events that lead to reduced muscle strength, however, remain unclear, complicating targeted disease management.

##### Anti-HMGCR IMNM

HMGCR is an enzyme located at the membrane of the endoplasmic reticulum, and is the pharmacologic target of statins [26]. Antibodies against HMGCR have been found in biopsy specimens of IMNM patients and much less frequently in other muscle conditions [80,127]. These antibodies are associated with statin exposure in approximately half of the cases [25]. Statin-unexposed anti-HMGCR patients had higher CK levels and were younger [128]. The exact pathogenesis is not known, but a genetic predisposition (class II HLA allele DRB1*11:01) and increased overexpression of HMG-CoA reductase in muscle tissue following exposure to statins both play a role (Mammen 2016). Regenerating muscle cells express high levels of HMG-CoA reductase protein [26,129], which is required for normal muscle-cell differentiation [130,131]. Additional genetic risk factors and environmental triggers are probably involved but largely unknown [132].

The absence or a low number of infiltrating lymphocytes and the presence of MAC on non-necrotic muscle-cell membranes suggest that anti-HMGCR autoantibodies may be pathogenic. This is supported by the observation that, similarly to anti-SRP antibodies, levels of anti-HMGCR antibodies correlate with disease activity (CK levels) and severity (muscle weakness) [27,127,128].

However, HMG-CoA reductase is not known to reside on the sarcolemma, where it could be targeted by autoantibodies. Alternatively, anti-HMG-CoA reductase autoantibodies might cross-react with a different, unidentified antigen. Whether autoantibodies, some other soluble factor, or infiltrating immune cells cause the muscle damage remains to be determined (Mammen 2016).

In the abovementioned in vivo study on the pathogenicity of anti-SRP autoantibodies, the role of anti-HMGCR autoantibodies was also investigated: similarly to anti-SRP autoantibodies, anti-HMGCR autoantibodies induced myotubular atrophy, increased pro-inflammatory cytokines, and impaired myoblast fusion due to decreased IL-4/IL-13 [55].

In an in vivo study mouse model, C57BL/6, Rag2 deficient (Rag2^−/−^), and complement component 3 deficient (C3^−/−^) mice were injected daily with plasma or purified antibodies from patients suffering from anti-HMGCR associated IMNM [119]. Muscle deficiency, evaluated by muscle strength on electrostimulation and grip test, tended to be less severe in mice receiving anti-HMGCR positive IgG compared to anti-SRP positive IgG. This seems to be consistent with observations in patients [112]. Histological analysis showed a statistically significant increase in necrotic myofibers after transfer of anti-HMGCR positive IgG [119].

## 4. Derangement of the Complement System in DM and IMNM

The complement system is a key part of innate immunity and modulates the adaptive immunity. There are three activation pathways: the classical, lectin, and alternative pathways. Surface-bound antibodies trigger the classical pathway, and amplification is accommodated by the alternative pathway, regardless of the initiating route of activation [133]. The classical complement pathway causes a cascade reaction of complement proteins, such as C1 and C3. Subsequently, C3 convertase cleaves C3 to produce C3b, which forms a complex with C4b and C2a. This complex is a C5 convertase, which cleaves C5 to produce C5a and C5b [134].

The lectin pathway starts with signal recognition by the oligomeric structures of mannose-binding lectin (MBL), ficolins and collectins, which activate mannan-binding lectin serine protease 1 (MASP1) and MASP2, which in their turn moderate production of C4b. Next, the lectin pathway follows the same steps as the classical pathway [134]. In the alternative pathway, C3 interacts with factor B and factor D, leading to cleavage of further C3. As the final step of this pathway, an additional C3b binds to the C3 convertase and forms a C5 convertase, which cleaves C5 [135]. All three pathways generate C5b, which triggers the second part of the system: the lytic pathway and formation of the MAC [136]. MAC, or C5b-9 complement complex, is the final step in the activation of the complement system. Complement deposition is not a unique feature of any IIM, as it may also occur (with divergent qualitative characteristics) in hereditary diseases of skeletal muscle [108].

### 4.1. Complement in DM

A comprehensive study in patients with DM examined the trigger for complement activation. Despite the absence of binding of immunoglobulin complexes (IgGs), the pattern of complement activation was consistent with activation of the classical pathway [137]. The MAC reactive capillaries and transverse vessels reacted for C1q, which therefore served as a recognition and regulatory protein for the classical pathway [36].

The description of typical dermatomyositis in patients with a hereditary deficiency of complement factor 2 (C2) illustrates that the classical and lectin pathways are not a conditio sine qua non for the development of DM, and that the alternative pathway may be primary involved in a subset of patients [138].

Plasma levels of activated complement factor 3 (C3a) and C5b-9/MAC in patients with DM with active disease were higher than in inactive DM and healthy controls, suggesting a role of complement in DM [139]. IVIG treatment showed a strong reduction in C3 uptake and disappearance of C3b and MAC in DM muscle biopsies, which was related to clinical improvement [140].

The presence of activated complement in DM is apparent in muscle biopsies. C4d staining in muscle biopsies of DM patients mirrored C5b-9 staining [141]. Both antibodies labeled the cytoplasm of degenerating necrotic myofibers. In addition, both antibodies showed distinct endomysial capillary labeling in a subset of dermatomyositis muscle biopsies including areas with perifascicular atrophy [141]. C5b-9/MAC deposition on capillaries is a specific finding in DM biopsies and the extent of C5b-9 deposits seems to be associated with the clinical course of the disease [36,142,143].

In very early stages of DM or in small muscle biopsies, inflammatory cell infiltrates may be lacking, while C5b-9 deposits on endomysial capillary vessel walls are present, which may precede the development of inflammatory cell infiltrates in DM. C5b-9 deposits are thought to be evidence for DM being caused by a microvasculopathy [144,145]. The pattern of capillary C5b-9 deposits is critical; non-DM patients may also show capillary complement deposits, but in a more diffuse endomysial pattern, as opposed to the perifascicular pattern in DM [146]. Although MAC formation is a typical feature in DM, the description of typical dermatomyositis in patients with hereditary deficiencies of complement factor 9 (C9) illustrates that DM may occur without the formation of C5b-9/MAC [147].

Despite uncertainties regarding the events that trigger complement activation and the relative contributions of each activation pathway to DM, there is evidence that DM is in part a complement-mediated microangiopathy, mostly mediated by the classical pathway [36].

### 4.2. Complement in IMNM

Although the mechanisms underlying myofiber necrosis have not been elucidated, given that IMNM patients have MAC on the sarcolemma of non-necrotic muscle fibers and that few or no cytotoxic T cells and/or natural killer cells are observed in muscle biopsies from patients with IMNM, myofiber toxicity may occur because of complement-mediated cell death that is antibody-dependent [27,44,45,107,109,148]. Activation of the classical complement cascade in seropositive IMNM patients has been identified by the presence of sarcolemmal IgG1 and C1q and by the formation of C5b–9/MAC [45,108].

The contribution of complement mediated muscle injury in IIM has been investigated in complement deficient and wild-type mice [119]. The authors showed that IgG from patients with seropositive IMNM was markedly less pathogenic in complement (C3)-deficient mice than in wild-type mice. Supplementing recipient mice with Ig-depleted pooled human complement serum augmented the in vivo pathogenicity of IgG from patients with seropositive IMNM [119]. Together, these data suggest an important role of the complement cascade in seropositive IMNM, although the exact contribution of complement pathways and factors needs to be further clarified [149].

## 5. Interferon Pathway

### 5.1. Interferon Overexpression in DM

The interferon (IFN) pathway can be activated by the binding of three different types of ligands to cell surface receptors: type 1 IFNs (i.e., IFN-α and IFN-β), type 2 IFNs (i.e., IFN-γ), and type 3 IFNs (i.e., IFN-λ) [150]. These proteins bind to their corresponding surface receptors and subsequently stimulate the expression of IFN-inducible genes via the Janus kinase (JAK)/signal transducer and activator of transcription (STAT) signaling pathway [151].

Myofibers in regions of PFA consistently express immunostaining of MxA, a IFN-1-induced protein, and MxA was also found in DM in normal-appearing perifascicular fibers [2]. Genes induced by IFN-1 are overexpressed in the muscle tissue of patients with DM. This overexpression is considered to be toxic to perifascicular muscle fibers and adjacent capillaries [59]. High levels of expression of IFN-1-inducible genes corresponded with different indicators of disease activity in DM, such as muscle weakness and elevated serum CK levels [152,153]. Considering the presence of interferon-inducible gene expression in patients with DM, Greenberg et al. investigated plasmacytoid dendritic cells (PDCs), known as interferon-producing cells, and effector cells of the innate immune system, in muscle biopsies (Greenberg, Pinkus et al., 2005) [154]. The finding of abundant PDCs indicated local production of interferon-α/β (IFN-1) [59].

In the innate immune response, the IFN signal activates the JAK-STAT intracellular cascade through interferon-stimulated gene (ISG)-15 conjugation to substrates [155]. ISG15 is a ubiquitin-like protein and its conjugation to substrates modifies their expression levels and activity [155].

A proteomic study showed that ISG15 binds to muscle proteins in DM muscles, such as MxA, suggesting that ISG15 could play a regulatory role in DM pathology [156].

Along with overexpression of ISG15 in muscle biopsies in DM with PFA, ISG15 expression levels alone can be used to accurately quantify the activation of the IFN-1 pathway in muscle biopsies in IIM [157,158]. Concerning the IFN-1-inducible genes, the quantity of overexpression was considerably different among different types of myositis and was most consistently shown in DM. Activation of the IFN pathway was associated with increased expression of inflammatory cell and muscle regeneration genes. REF(S)

Activation of the IFN-2 pathway was also seen in DM [158].

To elucidate signatures and gene networks that are associated with DM pathology, RNA-sequencing profiles from a large cohort of myositis patients were studied, which confirmed the overexpression of IFN-1 inducible genes in DM [159]. In blood, IFN-β but not IFN-α protein levels were found to be elevated in DM patients [160]. In a prospective cohort of 42 patients, the authors found that elevated serum protein levels of IFN-β but not IFN-α were present in DM, and these levels of IFN-β correlated with both type 1 IFN gene signature and skin disease activity, as reflected by the Cutaneous Dermatomyositis Disease Area and Severity Index (CDASI) [161].

### 5.2. Interferon Overexpression in IMNM

The IFN1 pathway is not only activated in patients with DM, but also in patients with IMNM, albeit to a lesser extent. While activation of the IFN2 pathway was seen in DM, this was not the case in IMNM [158]. Concerning the IFN1-inducible genes, the quantity of overexpression was considerably different among different types of myositis, being highest in DM and low in IMNM [158].

## 6. Pharmacological Treatment

A small group of patients with myositis responds very well to standard therapy and achieves complete remission after initial treatment (van de Vlekkert, Hoogendijk et al., 2014). However, for a significant proportion of patients with IIM, including DM and IMNM, chronic immunosuppressant therapy is needed because of ongoing disease activity. Despite this, disease damage (fatty infiltration of muscle tissue) may occur, leading to permanent disability [10,162]. Patients with subtypes of DM (anti-MDA-5) and IMNM (anti-SRP) often need more intensive treatment immediately following the diagnosis [2]. Below we first describe general treatment strategies for IIM, followed by descriptions of more intensive treatment strategies that are currently recommended for subtypes of IIM. Next, we discuss new targets for therapy based on pathophysiological insights (involvement of B cells and complement and interferon overexpression) followed by short descriptions of compounds targeting other pathways.

### 6.1. General Treatment Recommendations

The pharmacological treatment of IIM is mainly based on expert opinion/consensus [163]. No pharmacological treatment for IIM can be recommended based on randomized controlled trials (RCTs), except for intravenous immunoglobulin (IVIG) as an add-on treatment in refractory DM [164,165,166]. While formal proof from RCTs is lacking, current first-line therapy consists of glucocorticoids, e.g., high-dose prednisone (1–1.5 mg/kg/day), methylprednisolone or dexamethasone, for all IIM subtypes [162,167]. If possible, glucocorticoids should be tapered by 20–25% of the existing dose monthly with the goal of achieving a low daily dose of prednisone of approximately 5–10 mg daily within six months [162]. This non-targeted immunosuppressive therapy is efficacious, but is known for adverse effects and relatively high relapse rates. Very often, glucocorticoids are combined with steroid sparing agents such as methotrexate (MTX), azathioprine (AZA) or mycophenolate mofetil (MMF) for an additional immunosuppressive and prednisone-sparing effect [162,168]. As a prodrug of mycophenolic acid, treatment with MMF leads to immunosuppression by reducing proliferation of B cells and T cells. The effect of treatment with MMF is mostly limited to case series [169,170,171].

### 6.2. Intensive Treatment in Rapidly Progressive or Refractory DM and IMNM

#### 6.2.1. IVIG

IVIG modulates immune-mediated processes, and effectiveness in the treatment of myositis, among other antibody-mediated diseases, is broadly reported. Compared to glucocorticoids, IVIG is thought to have a relatively favorable safety profile and may have a prednisone sparing effect [171,172]. IVIG exerts pleiotropic effects, including action on B cells via different nonexclusive mechanisms, all leading to a reduction in activation [173]. In refractory DM, IVIG has been proven to be effective as an add-on treatment to glucocorticoids and may be added in patients with rapidly progressive and/or severe/refractory disease [164,165]. IVIG is also considered an effective treatment for refractory CADM based on two retrospective studies evaluating the clinical response of refractory cutaneous dermatomyositis in patients for whom IVIG was initiated specifically for skin disease [174,175]. Although relapses occurred, they were successfully treated with an additional course of IVIG [175].

Patients with IMNM have been successfully treated with IVIG monotherapy, although the number of reported patients is small [172,176]. As previously mentioned, IMNM is characterized by rapid onset and severe proximal muscle weakness, and aggressive early treatment should be considered. Early aggressive treatment with at least two immunotherapeutic agents and/or early treatment with IVIG can lead to favorable outcomes [177].

Our recent pilot study in treatment-naive patients with idiopathic inflammatory myopathies showed clinically relevant improvement in 45% out of 19 patients after six to nine weeks of first-line IVIG monotherapy [178]. Compared to most other immunomodulating compounds, IVIG acts more quickly, which could lead to early and sustained suppression of the inflammatory process when administered together with glucocorticoids. Considering the known effects of IVIG in IIM, both as add-on therapy in refractory patients and as monotherapy in newly diagnosed IIM, double-blind RCTs are needed to investigate the effect of add-on IVIG in patients with newly diagnosed IIM. Possibly early immunosuppression via intensive treatment (‘hit early, hit hard’) may induce faster reduction of disease activity and prevent chronic disability due to disease damage in myositis [179].

#### 6.2.2. Plasmapheresis

Plasmapheresis (PE) is used in various refractory autoimmune diseases in order to remove circulating autoantibodies and immune complexes [180]. Since IIM is associated with the presence of autoantibodies, a beneficial effect could be assumed, in particular in those subtypes with autoantibodies with a presumed pathogenic role (in particular anti-SRP and anti-HMGCR) as described above. However, a double-blind, placebo-controlled trial in refractory myositis failed to show efficacy [181]. A Japanese group investigated PE in anti-MDA-5 positive patients refractory to a combined immunosuppressive regimen, in a retrospective, single-center study. Patients who received PE showed a significantly better survival rate than those who did not. These results may suggest that PE is expected to be an effective adjuvant treatment in anti-MDA-5 positive patients, although larger prospective studies are needed [182].

#### 6.2.3. Cyclophosphamide 

Cyclophosphamide (CYC), an alkylating agent with strong cytotoxic and immunosuppressive effects, has been reported to be efficient in severe DM with progressive ILD [183]. In a large longitudinal cohort of 204 IIM patients, including 123 DM patients (60%), the treatment effect of oral CYC in refractory myositis was described [184]. Significant improvement in disease activity measures and decrease in daily prednisone dose were reported. Further RCTs are warranted to verify these results [184].

Because standard immunosuppressive, so-called non-targeted, therapies lead to considerable numbers of patients with inadequate response to treatment, numerous side effects, and insufficient control of disease, more targeted therapies are warranted [185]. Targeted (biological) therapies, which target immune cells or cytokines directly via an antibody or small molecule, may greatly improve outcomes in patients with IIM [171,186]. In terms of level of *evidence*, to the best of our knowledge, one RCT has investigated a biological in IIM, rituximab, which depletes B cells, and is described in more detail below [187].

## 7. Pharmacological Compounds Targeting B Cells, Interferon Pathway and Complement

### 7.1. B Cell Depletion

B cell-depleting rituximab is the most commonly used biological, and is mainly used in patients with refractory IIM. The effect of B cell depletion therapies (e.g., rituximab) supports the role of B cells in myositis pathogenesis [65,66,67,187]. While the previously mentioned RCT, called the Rituximab in Myositis (RIM) trial, failed to meet the primary outcome, 83% of all patients met the predefined definition of improvement [187]. The presence of specific MSAs might be associated with differences in treatment response. In a subgroup analysis of the RIM trial, the presence of anti-Mi-2 autoantibodies was a strong predictor (Hazard Ratio 2.5, *p* < 0.01) of clinical improvement after B cell depletion with rituximab [187,188].

A retrospective cohort study included 43 IIM patients with inadequate response to at least two immunosuppressive/immunomodulatory therapies who received two infusions of rituximab, one at baseline and one at six months, in order to maintain baseline immunosuppressive treatment. Clinical and laboratory improvement (defined as > 20% improvement in at least three of the six modified IMACS core set measures) was seen in all patients at six months and the improvement persisted at one year [66].

A recent systematic review including 17 articles concluded that rituximab showed good effect in the treatment of ILD related to anti-MDA-5 positive DM. Prior to treatment with rituximab, the majority (~80%) of patients were treated with medium- or high-dose glucocorticoids and at least one additional immunotherapeutic agent [189]. Therefore, rituximab could be a promising early treatment for anti-MDA-5 DM with ILD.

Belimumab is a humanized monoclonal antibody targeting the cytokine BAFF, which has been approved as a treatment for systemic lupus erythematosus (SLE) [190]. A double-blind placebo-controlled trial investigating the efficacy and safety of belimumab in patients with IIM with a history of inadequate response to three months of glucocorticoids and/or one other immunosuppressive agent is currently ongoing (ClinicalTrials.gov Identifier: NCT02347891).

### 7.2. Interferon Pathway

#### 7.2.1. Anti-IFN-α Antibody

Sifalimumab is an anti-IFN-α monoclonal antibody. In a double-blind, phase 1b clinical trial, 51 patients were enrolled, including 26 DM patients. To evaluate the pharmacodynamic effects of sifalimumab in blood and muscle of myositis patients, neutralization of a type I IFN gene signature (IFNGS) was measured. Sifalimumab suppressed the IFNGS in blood, and a similar trend existed in muscle biopsies, although weaker [191]. Patients with 15% or greater improvement on MMT scores showed greater neutralization of the IFNGS than patients with less than 15% improvement in both blood and muscle.

#### 7.2.2. Janus Kinases (JAK) Inhibitors

IFN-1 is suppressed by the JAK inhibitor ruxolitinib, and since the IFN-1 pathway plays a prominent role in the pathogenesis of refractory DM, its use in future treatment is of interest. The effect of IFN-1 suppression by ruxolitinib is seen in both muscle and in endothelial cells [192]. The authors reported a clinical improvement after 3 months of treatment in four patients with refractory DM with reduced serum IFN-1 levels. Results of a first proof-of-concept, open-label, prospective clinical trial of the JAK inhibitor tofacitinib in 10 patients with refractory DM, who had been treated with a minimum of a 12-week trial of steroid therapy and at least one other first-line immunosuppressive agent, showed efficacy, as measured by the ACR/EULAR myositis response criteria, and in particular in terms of skin changes [193].

### 7.3. Complement Inhibition

#### Anti-C5 Monoclonal Antibodies

The literature describes one patient with refractory dermatomyositis (with concomitant thrombotic microangiopathy) showing robust improvement of muscle weakness, edema, myalgia and CK levels following treatment with a C5-blocking agent (eculizumab) after several other treatments (steroids, IVIG, and PE) had failed [194]. In a randomized, double-blind, placebo-controlled 8-week pilot study in 13 DM patients, treatment with eculizumab showed improvement from baseline in terms of Manual Muscle Testing (MMT) (6%), physician global score (9%) and skin disease score (37%) in the treatment arm. The placebo arm showed worsening. No serious adverse effects were noted, and the incidences of minor adverse effects were the same in the treatment and placebo arms [195].

In a phase 2 RCT, 27 IMNM patients with anti-SRP or anti-HMGCR autoantibodies received either daily subcutaneuous zilucoplan, a complement C5 inhibitor, or placebo for 8 weeks. Zilucoplan did not yield a significant change in CK levels, which was the primary outcome. No unexpected safety findings and no relevant safety differences between zilucoplan and placebo were identified (ClinicalTrials.gov identifier: NCT04025632).

A double-blind, randomized, placebo controlled trial with ravulizumab, a humanized monoclonal complement C5 inhibitor, is ongoing. In this large trial of 180 adult patients with DM with inadequate response to two or more standard DM treatments, i.e., glucocorticoids or immunosuppressive/immunomodulatory therapies, intravenous ravulizumab is administered as a loading dose followed by maintenance doses every 8 weeks, with primary and secondary outcomes measured at 26 and 50 weeks respectively (ClinicalTrials.gov Identifier: NCT04999020).

## 8. New Pharmacological Compounds Targeting Other Pathways

### 8.1. Tumor Necrosis Factor (TNF)-α

Tumor necrosis factor (TNF)-α, a pro-inflammatory cytokine, is present in elevated levels in muscle biopsies in IIM patients [196]. Excessive production of TNF-α may play a role in the pathogenesis of myositis, and therefore may lead to a therapeutic role for TNF-α inhibitors, such as infliximab or etanercept, in the treatment of IIM [197]. A retrospective study in eight patients, six of whom were treated with etanercept, one with infliximab and one with both agents, showed improved motor strength and decreased fatigue [198]. In a pilot study, treatment with infliximab in refractory myositis, defined as failure to respond to treatment with high doses of glucocorticoids for a minimum of 6 months, failed to be effective [199]. An RCT with a cross-over period showed improvement in some patients treated with infliximab. Sample size was limited (*n* = 12), and only 4 out of 12 patients showed improvement in the second phase of the study at 40 weeks. Future studies with larger sample sizes are needed [200]. A randomized, double-blind, placebo-controlled trial of etanercept in DM patients showed that etanercept was well-tolerated and had a significant steroid-sparing effect [201]. The use of TNF inhibitors in IIM is debated at the moment, since recent studies report patients developing autoimmune diseases after treatment [202,203].

### 8.2. Interleukine (IL)-Receptor Antagonists

Interleukin-6 (IL-6), a pro-inflammatory cytokine, is expressed in the muscle biopsies of patients with different subtypes of myositis, including dermatomyositis [204]. The exact role IL-6 in the pathogenesis of DM needs to be clarified. After approval of the IL-6 receptor antagonist tocilizumab for treatment of rheumatoid arthritis, interest in its use for other autoimmune diseases, such as IIM, has increased. A recent randomized, double-blind controlled trial failed to show the efficacy of tocilizumab in adults with refractory dermatomyositis [205].

Anakinra is a recombinant IL-1 receptor antagonist that has been approved for the treatment of rheumatoid arthritis. Increased expression of IL-1α and IL-1β is seen in the muscle tissue of IIM patients [206]. Fifteen patients, refractory to high-dose prednisolone in combination with AZA or MTX, were included in an open-label case study and received a daily dose of 100 mg subcutaneous anakinra. Seven patients showed a clinical response, according to the International Myositis Assessment and Clinical Studies Group (IMACS) definition of improvement. Patients with more extramuscular symptoms, such as ILD, responded better. Blood analysis demonstrated that blocking of the IL-1 receptor reduced T cell differentiation [207]. A large, randomized controlled trial is needed to prove the efficacy and safety of anakinra in IIM.

### 8.3. Inhibition of T-Cell Costimulation

Abatacept prevents T-cell activation by binding to CD80 and CD86 on antigen-presenting cells [208]. A recent randomized open-label trial included 20 patients with myositis refractory to glucocorticoids and/or another immunosuppressant for at least 3 months. Nearly half of the patients showed improvement in muscle strength and health-related quality of life at 6 months [209]. A large phase III, double-blind, randomized trial evaluating the efficacy of subcutaneous abatacept, in addition to standard treatment, is currently ongoing (ClinicalTrials.gov identifier: NCT02971683).

## 9. Discussion

In summary, we reviewed recent advancements in the understanding of the pathophysiology and treatment of two of the most prevalent and severe subtypes of myositis, DM and IMNM. Based on new evidence related to the contributions of B cells, pathogenic antibodies, complement pathways and interferon overexpression, new therapeutic targets have been identified. The promising effects of targeted therapies on laboratory and clinical outcomes in small series of myositis patients need to be corroborated in ongoing and future clinical trials.

## Figures and Tables

**Figure 1 ijms-23-04301-f001:**
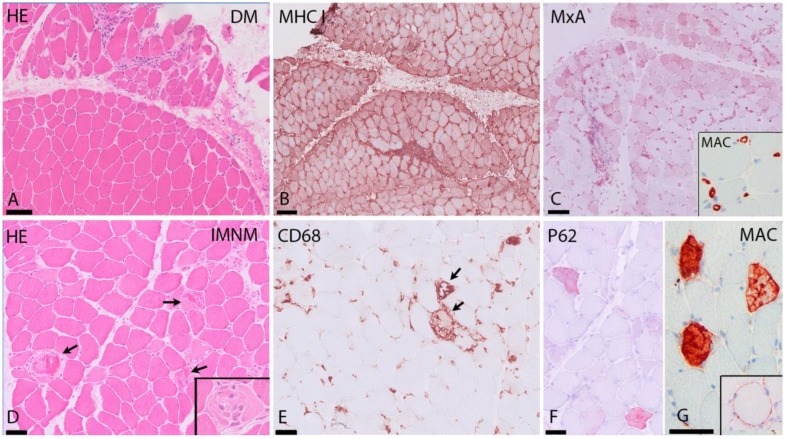
Typical findings in DM and IMNM. (**A**–**C**) DM. (**A**) (H&E staining) shows perifascicular atrophy in DM. (**B**) MHC I immunostaining shows strong sarcolemmal upregulation in the perifascicular area. (**C**) MxA immunostaining reveals strong expression on myofibers in the perifascicular area. Insert: MAC (complement C5b9) immunostaining with strong expression in capillaries. (**D**–**G**) IMNM. (**D**) (H&E staining): diffuse necrosis in several muscle fibers in SRP-IMNM (arrows and insert in (**A**)). (**E**) CD68+ cells in SRP-IMNM with myophagocytosis of necrotic fibers (arrows). (**F**) diffuse sarcoplasmic expression of p62 in scattered myofibers in IMNM. (**G**) MAC immunostaining showing sarcoplasmic and sarcolemmal (insert) MAC deposition in fibers. HE = hematoxylin and eosin stain, DM = dermatomyositis, MHC I = major histocompatibility complex I, MxA = myxovirus-resistance protein A, MAC = membrane attack complex, IMNM = immune-mediated necrotizing myopathy. Scale bar: (**A**–**C**) 100 μm; (**D**–**G**) 50 μm.

**Table 1 ijms-23-04301-t001:** Overlapping and diverging clinical characteristics of idiopathic inflammatory myopathies.

Characteristic	DM	IMNM	ASyS	OM
Muscle weakness	++	+++	++	++
Malignancies	++	+	−	−
Rapid progression *	++	+++	−	−
Raynaud’s	−	−	++	++
Skin involvement	+++	+	++	+
Interstitial lung disease	+	−	+++	++
Peri/myocarditis **	+	+	+	+

* in subtypes (e.g., IMNM with SRP autoantibodies and DM with MDA-5 antibodies (related to ILD)). ** described in case series of, predominantly, SRP-positive IMNM patients [19,20]. − = not present, + = rarely present, ++ = sometimes present, +++ = (very) often present. DM = dermatomyositis, IMNM = immune-mediated necrotizing myopathy, ASyS = anti-synthetase syndrome, OM = overlap myositis, SRP = signal recognition particle, MDA-5 = anti–melanoma differentiation gene 5, ILD = interstitial lung disease.

**Table 2 ijms-23-04301-t002:** Autoantibodies in dermatomyositis and immune-mediated necrotizing myopathies.

Diagnosis	Demographic Characteristics	Clinical Features	Autoantibodies	Other Features
Dermatomyositis (DM)	All ages Peak incidence 30–50 years 70% female	Symmetrical weakness Proximal > distal weakness Neck flexion > extension weakness Dysphagia Subacute onset DM-specific rash	MDA-5	Amyopathic/pauci-myopathic (CADM) Rapidly progressive severe ILD Classic and atypical rash, mechanic’s hands, palmar papules, ulceration of the fingers, panniculitis Polyarthritis
			NXP-2	Malignancies, such as breast and gastrointestinal cancer Severe weakness Calcinosis and ulceration
			Mi-2	Often severe weakness and highly elevated CK Classic DM-rash on sun-exposed skin Favorable long-term prognosis
			TIF1-γ	Malignancies; gastric, lung, breast, esophageal, bladder, colorectal, ovary, and thymus cancer. Pauci-myopathic Dysphagia Classic DM-rash
			SAE-1	Malignancy; adenocarcinoma from cervical, pulmonary, esophageal or rectal origin Amyopathic at onset: skin involvement before weakness Dysphagia ILD (Asia)
Immune-mediated necrotizing myopathy (IMNM)	All ages 64% female	Severe symmetrical weakness Axial weakness Muscle atrophy Dysphagia Highly elevated CK (>10× upper limit) Subacute onset Early fatty infiltration muscle-MRI	HMGCR	Use of statins Malignancy; no specific type predominant
			SRP	Severe weakness Therapy-resistant Early treatment with rituximab or intravenous immunoglobulins
			Seronegative	Malignancy; no specific type predominant Cardiac involvement

## Data Availability

Not applicable.

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
