# Peer review of "Pathophysiological Mechanisms and Treatment of Dermatomyositis and Immune Mediated Necrotizing Myopathies: A Focused Review"

_ijms, 2022, doi:10.3390/ijms23084301_

Round 1

Reviewer 1 Report

Kamperman et al. have written a very comprehensive review on dermatomyositis and immun necrotizing myopathy. It is well organized and comprises a lot of insights especially in molecular mechanisms of the diseases.

I only have some minor suggestions to improve the manuscript:

  1. first paragraph some commas missing
  2. page 4 concerning INNM due to COVID...also other forms of myositis are described due to COVID (see Aschman et al JAMA Neurol 2021) which should be mentioned, it is not mainly INNM.
  3. paragraph "pathophysiolog. mechanisms": a clear subdivision in DM and INNM is missing. I would recommend to keep the same order of the two diseases in each subsection.
  4. table with myositis specific antibodies: I would recommend to insert the type of malignancies associated with the antibodies. Why are only 2 of 5 dermatomyositis associated antibodies described in the text?
  5. IVIG is especially useful for refractory cutaneous disease manifestations in DM (see e.g. Femia AN, et al. J Am Acad Dermatol. 2013;69(4):654–7 and Bounfour et al. J Eur Acad Dermatol Venereol. 2014) which should be adressed in the text
  6. I would recommend a scetch/figure on where the old and new therapeutic agents work on the molecular level.

Reviewer 2 Report

In this review, the authors examine pathophysiological mechanisms, as well as current and upcoming therapeutic approaches in two common immune-mediated myopathies: dermatomyositis and immune-mediated necrotising myopathy. The review is well-written, comprehensive, and presents an up-to-date view of the field. I do however have a few comments and suggestions to make:

  1. Line 40: I would say that both IBM and IMNM are very muscle-predominant
  2. Line 59: I think this should say "along with..." rather than "among...".
  3. Table 1: There have now been a number of publications suggesting the presence of cardiac involvement in IMNM patients. The evidence presented is often imperfect, but I think it's nonetheless worth acknowledging in the table with a + or +/-.
  4. Line 105: necrosis is only very infrequently observed in DM biopsies (unlike antisynthetase syndrome). While many perifascicular fibres in DM have an appearance that is typically associated with regeneration, it is unlikely that these represent true regeneration after necrosis (as is seen in other myopathies).
  5. Line 122: I would mention in this paragraph capillary MAC deposition as a characteristic pathological finding in DM. This helpful for diagnostic purposes and is thought by to play a role in DM pathophysiology, as the authors describe further in the review.
  6. Line 123: A paragraph describing biopsy findings in IMNM would be helpful for readers, even though these findings are largely nonspecific.
  7. Lines 143-145: It's true that the exact trigger of IIM in many patients remains unclear and these cases are tentatively attributed to things like viral infections. There are nonetheless some patients in whom a clear trigger can be identified (e.g. statin exposure or malignancy). This should be more explicitly acknowledged here.
  8. Table 2: I would add that several studies have found an overall better long-term prognosis in anti-Mi-2 DM.
  9. Line 365: This line should start with "HMGCR is an enzyme..." rather than "Anti-HMGCR is an enzyme..." 
  10. Line 477: This should be corrected to "binding of three different types"
  11. Line 493: IFN may also be produced by other cell types in muscle, particularly fibroblasts.
  12. Line 510: It would be useful to discuss the question of IFNα vs IFNβ as the key driver of pathology in DM, as this has some therapeutic implications. IFNβ has been found to be more consistently elevated in DM and better correlates with disease activity (Liao et al, 2011; Huard et al, 2017).
  13. Line 534: There was also an RCT comparing prednisone alone to prednisone + methotrexate or cyclosporine in JDM, which showed that combination therapy is more effective than prednisone alone and that methotrexate is better tolerated than cyclosporine (Ruperto et al, Lancet. 2016;387(10019):671-678.).
  14. Line 558: Somewhere in this paragraph, I would highlight that IMNM is usually the most rapidly progressive IIM and associated with the most severe weakness, and thus warrants special consideration for early aggressive treatment in most patients. There is evidence that early use of 2+ immunosuppressants and early use of IVIG improve outcome in IMNM (Kassardjian et al, JAMA Neurol. 2015;72(9):996-1003).
  15. Line 630: I think this should be sifalimumab, not sufalimab. What has been published is that patients who had a >15% improvement on MMT over the course of the trial were more likely to also have demonstrated modulation of IFN-1 related genes. This trend was seen more clearly when gene expression was measured in blood, but less when it was measured in muscle, which in my mind casts some doubt on validity of the finding. I would thus not write that "Treatment with sifalimumab showed clinical improvement measured by manual muscle testing", which seems to imply an outright clinical improvement in the sifalimumab group compared to placebo (to my knowledge, this has not been claimed, unless there is some data of which I am unaware). My understanding is that development of sifalimumab has been discontinued.
  16. Line 642: This should be "refractory DM"

Reviewer 3 Report

The Authors report a comprehensive and updated review on immunopathogenesis, clinical features, diagnosis and treatment of immune mediated myopathies with particular emphasis on dermatomyositis and immune mediated necrotizing myopathies. This review may represent a useful summary on this topic for the daily practice of clinical immunology, neurology and rheumatology specialists.

1. The manuscript is an updated review  on immunopathogenesis, clinical features, diagnosis and treatment of immune mediated myopathies with particular emphasis on dermatomyositis and immune mediated necrotizing myopathies.

2. The topic is relevant in the field and addresses a relatively new issue that may help clinical immunology, neurology and rheumatology specialists.

3. The manuscript adds only few new infromation to the current literature

4. The manuscript is a review therefore there is not a methodology section

5. The conclusions are consistent with the arguments presented in the text and address the man questions posed

6. References are appropriate

7. Tables and fugures are appropriete 

Author Response

Plese see the attachment. 

Round 2

Reviewer 2 Report

The authors have adequately addressed all my suggestions. The manuscript is well-written and informative, and I think it is suitable for publication in its current form.